# ALAM: Averaged Low-Precision Activation for Memory-Efficient Training of Transformer Models

**Sunghyeon Woo, Soonwoo Lee, Dongsuk Jeon**
Seoul National University, Seoul, Korea
{wsh0917, ori915, djeon1}@snu.ac.kr

## Abstract

One of the key challenges in deep neural network training is the substantial amount of GPU memory required to store activations obtained in the forward pass. Various Activation-Compressed Training (ACT) schemes have been proposed to mitigate this issue; however, it is challenging to adopt those approaches in recent transformer-based large language models (LLMs), which experience significant performance drops when the activations are deeply compressed during training. In this paper, we introduce ALAM, a novel ACT framework that utilizes average quantization and a lightweight sensitivity calculation scheme, enabling large memory saving in LLMs while maintaining training performance. We first demonstrate that compressing activations into their group average values minimizes the gradient variance. Employing this property, we propose Average Quantization which provides high-quality deeply compressed activations with an effective precision of less than 1 bit and improved flexibility of precision allocation. In addition, we present a cost-effective yet accurate sensitivity calculation algorithm that solely relies on the L2 norm of parameter gradients, substantially reducing memory overhead due to sensitivity calculation. In experiments, the ALAM framework significantly reduces activation memory without compromising accuracy, achieving up to a $10\times$ compression rate in LLMs.

## 1 Introduction

In recent years, deep learning has demonstrated human-like or even better performance on language-related tasks using large language models (LLMs) (Chen et al., 2023; Smith et al., 2022; OpenAI, 2023; Touvron et al., 2023a;b). Behind this success lies various efforts to increase the model size since it directly improves model performance due to scaling laws Kaplan et al. (2020); Sorscher et al. (2022a;b). However, the amount of memory required for training also increases proportionally, hindering their practical implementation. One of the reasons behind the memory bottleneck during training is that the backpropagation algorithm (Kelley, 1960) needs to store all intermediate activations generated during the forward pass in memory for later use in the backward pass for calculating parameter gradients, which leads to extensive memory requirements. For instance, as shown in Fig. 1, GPT-3 (Chen et al., 2023) with 175B parameters and MT-LNG (Smith et al., 2022) with 1T parameters require 67.3 GB and 132.7GB activation memory, respectively, exceeding the memory occupied by the parameters and optimizer state (Korthikanti et al., 2022). This issue even worsens with larger micro-batch size or sequence length because the activation memory also increases proportionally whereas the memory occupied by the parameters and optimizer state remains unchanged. Therefore, it is crucial to reduce activation memory when it comes to training LLMs.

Activation rematerialization (Chen et al., 2016; Jain et al., 2019; Feng & Huang, 2021) and reversible networks (Gomez et al., 2017; Kitaev et al., 2020; Sander et al., 2021; Cai et al., 2023) store only part of the activations, recomputing the rest during the backward pass. These methods demand extra computation in the backward pass and reduce training speed. Reduced-precision training (Micikevicius et al., 2018; Wang et al., 2018; Chen et al., 2020; Sun et al., 2020) aims at reducing computation precision and training memory by representing each variable in training (e.g., weight, error, activation, and gradient) with low-precision data formats such as FP8. However, training accuracy drops

Figure 1: Memory usage breakdown of GPT-3 (22B/175B) and MT-LNG (530B/1T) with data and model parallelism. The dashed red line denotes the 80GB capacity of the NVIDIA A100 GPU.

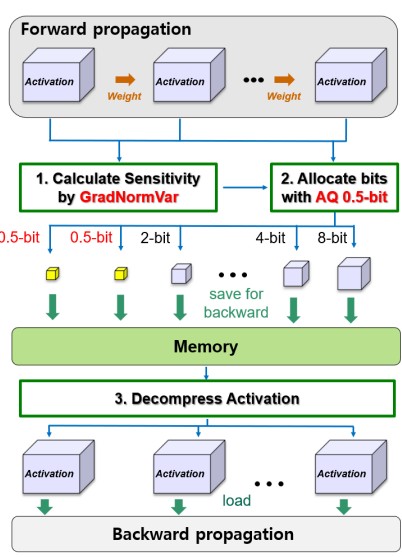

Figure 2: ALAM training framework.

quickly with lower precision, and optimized kernels for low-precision operations are required to fully utilize its advantage in training speed. On the other hand, Activation-Compressed Training (ACT) (Chakrabarti & Moseley, 2019; Chen et al., 2021; Liu et al., 2022b; Pan et al., 2021; Liu et al., 2022a) solely aims to reduce activation memory usage by compressing the activations before storing them in the forward pass. Conventional ACT methods, however, exhibit large performance drops for high compression rates when applied to LLMs. For example, MESA (Pan et al., 2021) compresses all layers uniformly with an identical compression rate, noticeably undermining training performance for activation precision under 8 bits. GACT (Liu et al., 2022a) determines compression rates (i.e., allocated bit precision) of each layer based on its sensitivity, achieving a state-of-the-art compression rate of 4 bits in transformer models. However, deeply quantized activations noticeably affect training performance, and the lowest bit precision it can allocate to less-sensitive layers is 1 bit, which limits the flexibility of bit precision allocation and hinders further compression.

In this work, we propose a new ACT framework, ALAM, which greatly improves compression rates while maintaining training performance. We first propose to represent a group of activations with their group average value as a means to compress them deeply and mathematically show that this scheme minimizes gradient variance due to activation compression. Based on this observation, we propose Average Quantization (AQ), which replaces a group of activations with their group average to realize sub-1b precision effectively. We experimentally demonstrate that the sub-1b activations obtained by AQ maintain high quality, even surpassing that of conventional 1-bit quantized activations. Moreover, by assigning sub-1b precision to the layers with very low sensitivity, we can further enhance the overall compression rates. We also introduce the GradNormVar algorithm that determines the sensitivity of a layer based on the variance of the L2 norm of parameter gradients. This approach significantly reduces the memory required for sensitivity calculation, when compared to GACT (Liu et al., 2022a) which requires the variance of parameter gradients from multiple seeds. Experimental results confirm that our ALAM outperforms the state-of-the-art method in various transformer models including LLaMA2-7B/13B, and LLaMA-30B. In summary, the main contributions of our paper are:

- We propose Average Quantization, which generates high-quality sub-1b activations, enabling further compression without sacrificing accuracy. To the best of our knowledge, this is the first attempt to compress activations through simple averaging.

- We propose GradNormVar, a lightweight sensitivity calculation algorithm that employs the variance of the L2 norm of parameter gradients, eliminating the need to retain all parameter gradients and substantially reducing memory usage.

- The proposed ALAM framework is demonstrated to successfully compress activations in various transformer models, achieving a $9.9\times$ compression rate in LLaMA-30B.

## 2 ACTIVATION-COMPRESSED TRAINING

### 2.1 FORMULATION

We represent the loss function of forward propagation in a neural network as $\mathcal{L}(\theta, X_0)$, given input $X_0$ and parameters $\theta$. When the stochastic gradient descent (Ruder, 2016) is applied, the parameters are updated as $\theta^{t+1} = \theta^t - \eta \nabla_{\theta^t} \mathcal{L}(\theta^t, X_0)$, where $t$ denotes the iteration number and $\eta$ is the learning rate. Then, the gradients are calculated as follows:

$$\nabla_{X_{l-1}}\mathcal{L}, \ \ \nabla_{\theta_l}\mathcal{L} = G_l(\nabla_{X_l}\mathcal{L}, \theta_l, X_{l-1}) \tag{1}$$

Here, $X_l$ and $\theta_l$ denote the activation and parameter of the $l$-th layer, respectively. $G_l$ denotes the gradient function of the $l$-th layer, generating the input gradients $\nabla_{X_{l-1}}\mathcal{L}$ and parameter gradients $\nabla_{\theta_l}\mathcal{L}$ from the output gradient $\nabla_{X_l}\mathcal{L}$, parameters $\theta_l$, and activations $X_{l-1}$. For the sake of simplicity, we denote $\mathcal{L}(\theta, X_0)$ as $\mathcal{L}$. Given Eq. 1, all layer activations $X_{l-1}$ must be stored in memory during the forward pass to compute gradients, leading to substantial memory use. The ACT frameworks are designed to tackle this memory consumption issue by compressing activations as follows:

$$\hat{\nabla}_{X_{l-1}}\mathcal{L}, \ \ \hat{\nabla}_{\theta_l}\mathcal{L} = G_l(\hat{\nabla}_{X_l}\mathcal{L}, \theta_l, \hat{X}_{l-1}) \tag{2}$$

where $\hat{X}$ represents the compressed activation, and $\hat{\nabla}$ represents the gradients obtained by ACT. For instance, if we compress activations from FP32 to INT1, a $32\times$ reduction in activation memory usage can be achieved.

### 2.2 CONVERGENCE OF ACTIVATION-COMPRESSED TRAINING

If $\mathcal{L}$ is continuously differentiable, $\nabla \mathcal{L}_\theta$ is $L$-Lipschitz for some constant $L$, $\mathcal{L}$ is bounded below by $\mathcal{L}_{inf}$, and there exists $\sigma^2 \geq 0$ satisfying $\text{Var}\big[\hat{\nabla}_\theta\mathcal{L}\big] \leq \sigma^2$ for all parameters, then we have the following convergence theorem (Chen et al., 2021):

$$\mathbb{E}\big[\hat{\nabla}_\theta\mathcal{L}\big] \leq \frac{2(L - \mathcal{L}_{inf})}{\alpha t} + \alpha L \sigma^2 \tag{3}$$

As the iteration $t$ increases, the first term of the bound in Eq. 3 decreases to zero. Therefore, minimizing the gradient variance $\text{Var}\big[\hat{\nabla}_\theta\mathcal{L}\big]$ is critical for the convergence of ACT. For a stochastic quantizer, Liu et al. (2022a) demonstrated that the gradient variance can be expressed as:

$$\text{Var}\big[\hat{\nabla}_\theta\mathcal{L}\big] \leq \sum_{l=1}^{L} c_l (2^{b_l} - 1)^{-2} \tag{4}$$

where $b_l$ denotes the bit precision (i.e., the number of bits to present activation) of the $l$-th layer. The sensitivity $c_l$ is defined as how sensitive it is to activation compression. If the sensitivity is high, the gradient variance is larger at the identical bit precision, undermining training performance and requiring higher bit precision to mitigate it. Therefore, we can determine the optimal bit precision of each layer based on its sensitivity to maximize memory savings with minimal training performance degradation by reducing gradient variance in Eq. 4 using a greedy algorithm.

GACT (Liu et al., 2022a) calculates sensitivity $c_l$ by observing the variance of parameter gradients from multiple runs of actual training using different seeds for stochastic rounding. More specifically, it initially compresses activations using a specific seed and calculates parameter gradients. Then, the algorithm changes only the seed for compressing the $l$-th layer's activations and recalculates parameter gradients. Finally, it treats the variance of all parameter gradients between these seeds as the sensitivity of the $l$-th layer, indicating how changes in the $l$-th layer's activation impact gradient variance. This process is detailed in Algorithm 1.

---

**Algorithm 1** Sensitivity calculation in GACT

---

**Require:**
$g_i^{(l)}$: parameter gradient of $i$-th element in $l$-th layer.
$r_1, r_2$: two random seeds
$L$: number of layers, $n_l$: elements in $l$-th layer
$b = (b_l)_{l=1}^L$: compression scheme
**Ensure:**
$s^{(l)}$: sensitivity of $l$-th layer
    **for** $l = 1$ to $L$ **do**
        Set $r_1$ for compressing all activation $X$, run backprop and store $h_0 \leftarrow \{g_i^{(l)} | i \leq n_l, l \leq L\}$
        Change $r_1$ to $r_2$ only for compressing $l$-th layer's activation $x_l$, recalculate $h_1 \leftarrow \{g_i^{(l)} | i \leq n_l, l \leq L\}$
        $s^{(l)} = \frac{1}{2}\|h_0 - h_1\|^2 (2^{b_l} - 1)^2$
    **end for**

---

## 3   AVERAGE QUANTIZATION

### 3.1   MINIMIZING GRADIENT VARIANCE

In this section, we propose a new approach to activation compression and show that it minimizes the gradient variance in ACT. Chen et al. (2021) proved that the gradient obtained by ACT is unbiased (i.e., $\mathbb{E}[\hat{\nabla}_\theta] = \mathbb{E}[\nabla_\theta]$). Following this property, the gradient variance of ACT can be expressed as:

$$\text{Var}[\hat{\nabla}_\theta \mathcal{L}] = \text{Var}[\nabla_\theta \mathcal{L}] + \mathbb{E}[\|\Delta \nabla_\theta \mathcal{L}\|^2] \tag{5}$$

where $\Delta \nabla_\theta \mathcal{L} = \hat{\nabla}_\theta \mathcal{L} - \nabla_\theta \mathcal{L}$ . The details can be found in Appendix A. On the right-hand side of Eq. 5, the first term is the gradient variance of SGD, which is not related to activation compression. Consequently, the second term, representing the gradient difference caused by activation compression, plays an important role in minimizing the gradient variance in ACT. Evans & Aamodt (2021) derived boundary functions for gradient differences in commonly used layers in neural networks, such as fully-connected, convolution, batch normalization, and layer normalization layers. We further extend these boundary functions in Appendix A and prove that the boundary function of gradient difference can be simply expressed as

$$\|\Delta \nabla_\theta f(\theta, X)\|^2 \leq d\|\Delta X\|^2 \tag{6}$$

where $\Delta X = \hat{X} - X$, and $d$ is a positive constant. Now suppose we divide activations into several groups and represent each group using a single representative value, where its optimal value can be determined using the theorem below.

**Theorem 1.** *The value of the boundary function of gradient difference is minimized when each group of activations is approximately represented using its group average value.*

Theorem 1 can be proven by showing that the derivative of $d\|\Delta X\|^2$ with respect to an approximate value is equal to 0 when using average as the approximate value because $D(\Delta X)$ is a convex function with respect to the approximate value. Detailed proof can be found in Appendix A. Furthermore, we experimentally demonstrate this property by training a 512-512-10 MLP (Haykin, 1994) model on MNIST (Deng, 2012). We approximate each group of activations using various values: average, median, minimum, maximum, and randomly selected activation in a group, where we group adjacent elements in the flattened activation vector, and the group size is set to 32. Other experimental details are provided in Appendix D. Experimental results in Table 1 confirm that averaging is the most effective compression method, which achieves high training accuracy close to full-precision training.

Table 1: Test accuracy of MLP trained on MNIST with different activation approximation methods.

|  | FP32 | Average | Median | Minimum | Maximum | Random |
|---|---|---|---|---|---|---|
| Accuracy (%) | **98.72** | **96.09** | 84.82 | *fail* | *fail* | 90.31 |

## 3.2 AVERAGE QUANTIZATION

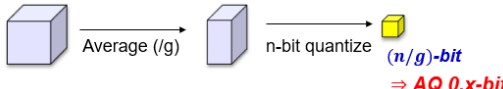

Figure 3: Average Quantization for high-quality sub-1b precision.

In Section 3.1, we show that averaging is an effective method to compress activation with minimal impact on training performance. By adopting this property, we propose Average Quantization, which generates high-quality compressed activation with sub-1b precision. We split activations into multiple groups and only store a single activation for each group, which is an average value of all the elements in the group. For instance, if we take an average for a group of four activations and quantize it to 2 bits, we only need to store 2 bits for a group. This achieves an identical compression rate to when we compress all of four activations separately to 0.5-bit precision in conventional activation compression, and hence we denote this as AQ 0.5-bit. Note that our Average Quantization flattens and groups adjacent activation elements during the forward pass, and then recovers them by repeating the group average in the backward pass without requiring an additional mask or spatial index for recovery.

We experimentally show that our approach produces high-quality compressed activation. In experiments, we train VGG-11 (Simonyan & Zisserman, 2015) on CIFAR-100 (Krizhevsky et al., 2009) using various types of compressed activations and compare test accuracy and validation loss. Fig. 4 displays the results for 2-bit uniform quantization, 1-bit uniform quantization, AQ 1-bit, and AQ 0.5-bit. Here AQ 1-bit and AQ 0.5-bit activations are obtained by averaging a group of two and four activations, respectively, and then quantizing the average value into 2 bits. Experimental details are provided in Appendix D. 1-bit uniform quantization fails to train the network, but AQ 1-bit succeeds in training, closely matching 2-bit uniform quantization. Even AQ 0.5-bit successfully trains the network without divergence, outperforming 1-bit uniform quantization. Therefore, we expect to replace 1-bit quantization with AQ 0.5-bit without sacrificing accuracy and allocate more bits to high-sensitivity activations for a given memory budget to improve training performance.

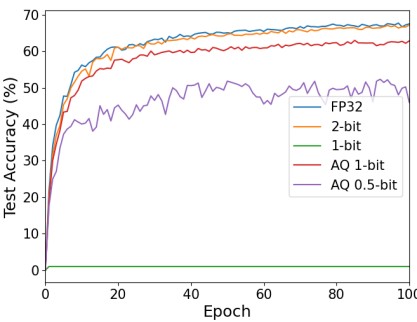

Figure 4: Comparisons of various compressed activations in VGG-11 training on CIFAR-100. For a fair comparison, activations of linear, convolutional, and batch normalization layers in VGG-11 are uniformly compressed in all experiments.

## 4 EFFICIENT SENSITIVITY CALCULATION

In ACT frameworks, it is crucial to accurately determine the sensitivity of a layer, as it greatly affects training performance. As discussed in Section 2.2 and Algorithm 1, GACT (Liu et al., 2022a) determines sensitivity by calculating the gradient variance from multiple runs of actual training with compressed activations using different seeds, and hence it must temporarily store all parameter gradients for each seed. Consequently, the peak memory usage substantially increases when calculating the sensitivity in models with a large number of parameters. For instance, GACT requires 700 GB memory for calculating sensitivity from two seeds when training GPT-3 with 175B parameters. To alleviate this issue, we propose the GradNormVar algorithm, which determines the sensitivity of layers as the variance of L2 norm of parameter gradients obtained from multiple seeds, as detailed in Algorithm 2. More specifically, our algorithm starts by compressing activations using a random seed. Instead of storing all parameter gradients, the algorithm calculates and retains only the L2 norm of parameter gradients for each layer. Subsequently, a different seed is used for compressing the $l$-th layer's activations, and the L2 norm of parameter gradients is recalculated. Finally, the algorithm employs the variance in the L2 norm of parameter gradients between these seeds as sensitivity

of the $l$-th layer. Since GradNormVar only stores the L2 norm of parameter gradients for each layer, rather than all parameter gradients from multiple seeds, it can significantly reduce the amount of memory required for calculating sensitivity.

---

**Algorithm 2** Proposed GradNormVar algorithm

---

**Require:**
  $G^{(l)}$: L2 norm of parameter gradients $g^{(l)}$ in $l$-th layer.
  $r_1, r_2$: two random seeds
  $L$: number of layers, $n_l$: elements in $l$-th layer
  $b = (b_l)_{l=1}^L$: compression scheme
**Ensure:**
  $s^{(l)}$: sensitivity of $l$-th layer
  **for** $l = 1$ to $L$ **do**
    Set $r_1$ for compressing all activations $X$, run backprop and store $h_0 \leftarrow \{G^{(l)} = \|g^{(l)}\|_2 | l \leq L\}$
    Change $r_1$ to $r_2$ only for compressing $l$-th activation $x_l$, recalculate $h_1 \leftarrow \{G^{(l)} = \|g^{(l)}\|_2 | l \leq L\}$
    $s^{(l)} = \frac{1}{2}\|h_0 - h_1\|^2(2^{b_l} - 1)^2$
  **end for**

---

To validate the effectiveness of our GradNormVar algorithm, we evaluated the sensitivity and bit precision allocation using the conventional method in GACT that employs gradient variance and our algorithm during the fine-tuning of BERT-Large (Devlin et al., 2019) on the MRPC dataset (Wang et al., 2019). As illustrated in Fig. 5, the sensitivity determined by our algorithm represents a similar trend to that of the conventional approach with a correlation of 0.9981, leading to comparable bit allocation. This is consistent with actual training performance; our GradNormVar algorithm achieves 86.9% training accuracy, closely matching the performance of conventional sensitivity calculation scheme in GACT (86.5%).

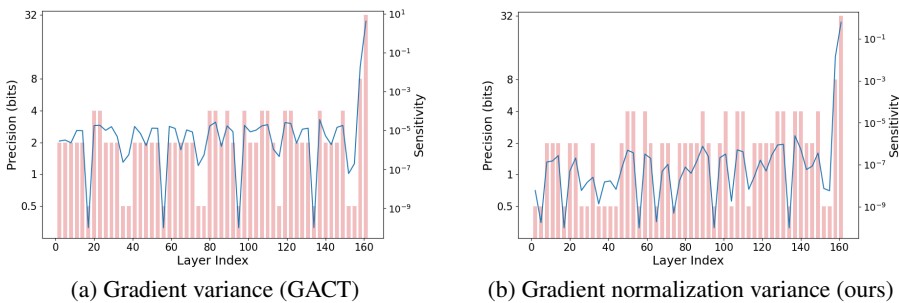

(a) Gradient variance (GACT)  (b) Gradient normalization variance (ours)

Figure 5: Sensitivity and bit preicision allocation comparisons when the target average precision is 2 bits in BERT-Large fine-tuning on MRPC datasets. Note that we adopt AQ 0.5-bit instead of conventional 1-bit quantized activation in both experiments.

## 5  ALAM: AVERAGED LOW-PRECISION ACTIVATION FOR MEMORY-EFFICIENT TRAINING

Combining the proposed techniques, we introduce ALAM, a new ACT framework that achieves a significant amount of memory saving while maintaining training accuracy. When the user specifies the target average precision, which represents the average precision of activations across all layers, the proposed ALAM framework automatically carries out ACT using the Average Quantization and GradNormVar algorithms. The detailed process is as follows:

1. **(Calculate sensitivity)** The sensitivity of each layer is calculated using the GradNormVar algorithm.

2. **(Allocate bit precision)** The bits are allocated to layers based on their sensitivity in a way that it minimizes the gradient variance in Eq. 4 using a greedy algorithm. AQ 0.5-bit is employed in low-sensitivity layers instead of conventional 1-bit quantization.

3. **(Training)** During the forward pass, activations are compressed based on the allocated bit precision, and then decompressed in the backward pass to update parameters.

Note that calculating sensitivity and allocating bit precision occur only before training and after 10% of total training epochs because activation sensitivity saturates early (Liu et al., 2022a).

## 6 RESULTS

In this section, we evaluate our ALAM framework by training diverse transformer models on various tasks. More details on the experiments can be found in Appendix D.

**Text Classification** We fine-tuned BERT models with different sizes, including DistilBERT-Base (Sanh et al., 2019), BERT-Base, and BERT-Large (Devlin et al., 2019), on the datasets in the GLUE benchmark (Wang et al., 2019) using three approaches: without activation compression (baseline), with GACT, and with ALAM. We evaluate test accuracy, activation memory, and the memory required for calculating sensitivity at various target average precisions for activations. Note that GACT and ALAM only compress float tensors, while they do not compress the other values such as masks and indices. Therefore, reducing the target average precision from 32 bits to 2 bits does not always result in a $16\times$ compression of activation memory, and the actual compression rate can vary by model. Table 2 shows that our ALAM achieves a significantly better trade-off between accuracy and memory savings compared to GACT in all cases. In DistilBERT-Base, ALAM achieves a compression rate of up to $16.4\times$ with <0.5 accuracy degradation in the WNLI dataset. Improvements from GACT expand further when fine-tuning BERT-Base on more challenging datasets such as CoLA, closely matching the uncompressed baseline in terms of accuracy. In BERT-Large, ALAM achieves activation memory savings of up to $22.5\times$ with accuracy comparable to the baseline. This improvement stems from the enhanced compression capability of ALAM at low-sensitivity activations.

Fig. 6 compares the sensitivity and bit precision allocation obtained by GACT and ALAM for BERT-Base on the MPRC dataset when the target average precision is set to 1 bit. GACT uniformly assigns 1-bit precision to all layers regardless of their sensitivity as the lowest precision supported in GACT is 1 bit, resulting in training failure. In contrast, ALAM allocates bit precision to layers differently based on their sensitivity, successfully training the model with 84.8% accuracy. Furthermore, while

Table 2: Test accuracy, activation memory (Act mem) with compression rate, and memory for calculating sensitivity (sens mem) in fine-tuning diverse language models on the datasets in the GLUE benchmark. The bolded text represents the test cases with < 0.5 score degradation compared to the non-compressed baseline, as well as the largest memory saving among the candidate algorithms with < 0.5 score degradation.

| Model | Dataset | Metric | Baseline 32-bit | GACT | | | ALAM (Ours) | | |
|---|---|---|---|---|---|---|---|---|---|
| | | | | 2-bit | 1.5-bit | 1-bit | 2-bit | 1.5-bit | 1-bit |
| Distil BERT-Base (66M) | STS-B | Corr | **87.1** | 86.5 | 81.2 | *fail* | **86.7** | 85.6 | 84.6 |
| | | Act mem | 834 MB | 60 MB (13.9×) | 48 MB (17.4×) | *fail* | **64 MB (13.0×)** | 50 MB (16.7×) | 36 MB (23.2×) |
| | | Sens mem | 0 | 766 MB | 753 MB | *fail* | 1.2 kB | 1.2 kB | 1.2 kB |
| | WNLI | Acc | **56.3** | **55.8** | 54.4 | 49.8 | **56.3** | 56.1 | 55.5 |
| | | Act mem | 834 MB | 63 MB (13.2×) | 48 MB (17.4×) | 35 MB (23.8×) | 64 MB (13.0×) | **51 MB (16.4×)** | 36MB (23.2×) |
| | | Sens mem | 0 | 766 MB | 753 MB | 753 MB | 1.2 kB | 1.2 kB | 1.2 kB |
| BERT-Base (110M) | CoLA | Corr | **59.3** | 56.0 | *fail* | *fail* | **59.1** | 55.8 | 54.5 |
| | | Act mem | 1678 MB | 130 MB (12.9×) | *fail* | *fail* | **135 MB (12.4×)** | 105 MB (16×) | 72 MB (23.3×) |
| | | Sens mem | 0 | 1240 MB | *fail* | *fail* | 2.3 kB | 2.3 kB | 2.3 kB |
| | RTE | Acc | **68.1** | 66.8 | *fail* | *fail* | **68.5** | 67.8 | 66.3 |
| | | Act mem | 1678 MB | 130 MB (12.9×) | *fail* | *fail* | 135 MB (12.4×) | **105 MB (16.0×)** | 72 MB (23.3×) |
| | | Sens mem | 0 | 1240 MB | *fail* | *fail* | 2.3 kB | 2.3 kB | 2.3 kB |
| | MRPC | Acc | **86.8** | 85.4 | *fail* | *fail* | **86.9** | 85.8 | 84.3 |
| | | Act mem | 1678 MB | 130 MB (12.9×) | *fail* | *fail* | **132 MB (12.7×)** | 103 MB (16.3×) | 75 MB (22.4×) |
| | | Sens mem | 0 | 1240 MB | *fail* | *fail* | 2.3 kB | 2.3 kB | 2.3 kB |
| BERT-Large (345M) | SST-2 | Acc | **93.0** | **93.1** | *fail* | *fail* | 93.2 | 93.4 | 92.8 |
| | | Act mem | 4434 MB | 360 MB (12.3×) | *fail* | *fail* | 368 MB (12.0×) | 282 MB (15.7×) | **197 MB (22.5×)** |
| | | Sens mem | 0 | 3817 MB | *fail* | *fail* | 4.6 kB | 4.6 kB | 4.6 kB |
| | QNLI | Acc | **92.4** | 91.5 | *fail* | *fail* | **92.1** | 92.1 | 90.7 |
| | | Act mem | 4434 MB | 358 MB (12.4×) | *fail* | *fail* | 366 MB (12.1×) | **279 MB (15.9×)** | 195 MB (22.7×) |
| | | Sens mem | 0 | 3817 MB | *fail* | *fail* | 4.6 kB | 4.6 kB | 4.6 kB |
| | MNLI | Acc | **86.5** | 85.2 | *fail* | *fail* | **86.4** | 85.8 | 84.8 |
| | | Act mem | 4434 MB | 360 MB (12.3×) | *fail* | *fail* | **371 MB (12.0×)** | 279 MB (15.9×) | 195 MB (22.7×) |
| | | Sens mem | 0 | 3817 MB | *fail* | *fail* | 4.6 kB | 4.6 kB | 4.6 kB |
| | QQP | Acc | **91.5** | 90.1 | *fail* | *fail* | **91.1** | 90.5 | 86 |
| | | Act mem | 4434 MB | 363 MB (12.2×) | *fail* | *fail* | **359 MB (12.4×)** | 282 MB (15.7x) | 197 MB (22.5×) |
| | | Sens mem | 0 | 3817 MB | *fail* | *fail* | 4.6 kB | 4.6 kB | 4.6 kB |

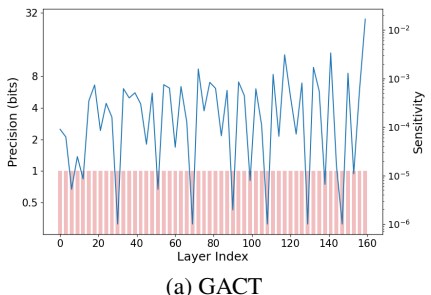 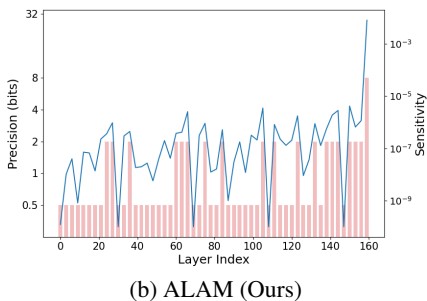

(a) GACT            (b) ALAM (Ours)

Figure 6: Sensitivity and bit precision allocation of GACT and ALAM when target average precision is 1 bit in BERT-Base fine-tuning on the MRPC dataset.

GACT requires a substantial amount of memory for calculating sensitivity (e.g., 3817 MB on BERT-Large), ALAM consumes only a negligible amount of memory, less than 4.6 kB in all experiments.

**Large Language Models** For evaluations using LLMs, we fine-tuned LLaMA2-7B (Touvron et al., 2023b) on the Alpaca dataset (Taori et al., 2023), and LLaMA2-13B (Touvron et al., 2023b) and LLaMA-30B (Touvron et al., 2023a) on the LIMA dataset (Zhou et al., 2023). Then, we observed 5-shot accuracy for MMLU (Hendrycks et al., 2021) and 0-shot accuracy for common sense reasoning (Bhakthavatsalam et al., 2021; Bisk et al., 2020; Zellers et al., 2019; Win, 2019), reading comprehension (Clark et al., 2019), and truthfulness tasks (Lin et al., 2021). In experiments, we applied parameter-efficient fine-tuning (PEFT) such as LoRA (Hu et al., 2022) and QLoRA (Dettmers et al., 2023). Table 3 shows ALAM achieves a similar accuracy to the baseline at 1-bit precision and provides up to $10.6\times$ compression rate, while GACT diverges even at 3-bit preci-

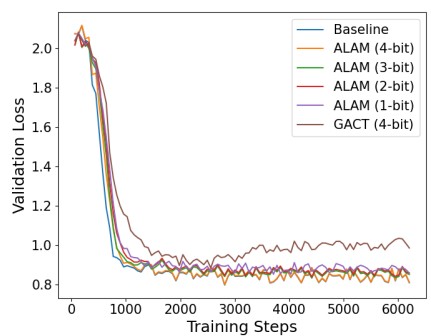

Figure 7: Learning curves when fine-tuning LLaMA2-7B on the Alpaca dataset.

sion. Our ALAM (1-bit) compresses activation by $10.3\times$ in LLaMA2-13B and $9.9\times$ in LLaMA-30B while closely matching the baseline in accuracy for all tasks. ALAM also significantly reduces memory for sensitivity calculations; however, the savings are not substantial because PEFT requires only a small number of parameter gradients. Fig. 7 displays learning curves obtained during the fine-tuning of LLaMA2-7B, demonstrating that our ALAM closely matches the baseline even at 1-bit precision while 4-bit GACT does not converge to the loss as rapidly as the baseline. Furthermore, the model fine-tuned with ALAM (1-bit) generates appropriate responses to the given prompt, as shown in Appendix B.

Table 3: Test accuracy, activation memory (Act mem) with compression rate, and memory for calculating sensitivity (sens mem) in fine-tuning LLMs. The bolded text represents the highest score and smallest activation memory while successfully training without divergence. The micro-batch size is set to 8 for LLaMA2-7B and 2 for LLaMA2-13B and LLaMA-30B.

| Model | PEFT | Scheme | Precision | Act mem | Sens mem | MMLU | Arc-c | PIQA | Hellaswag | WinoGrande | BoolQ | TruthfulQA |
|---|---|---|---|---|---|---|---|---|---|---|---|---|
| LLaMA2 -7B | LoRA | | Baseline | 16-bit | 21.1 GB | 0 | 46 | 47.6 | 79.5 | **75.7** | 68.8 | **77.9** | 41.8 |
| | | GACT | 4-bit | 5.9 GB (3.6×) | 24 MB | 44.8 | 46.6 | 78.9 | 75.5 | 68.2 | 77.1 | 40.4 |
| | | | 3-bit | 4.8 GB (4.4×) | 24 MB | *fail* | *fail* | *fail* | *fail* | *fail* | *fail* | *fail* |
| | | ALAM | 4-bit | 5.9 GB (3.6×) | 0.4 kB | **46.2** | **47.9** | 79.5 | 75.6 | 69.3 | 77.5 | 42.1 |
| | | | 3-bit | 4.8 GB (4.4×) | 0.4 kB | 45.2 | 47.6 | **79.8** | 75.4 | **69.5** | 77.2 | **42.4** |
| | | | 2-bit | 3.4 GB (6.2×) | 0.4 kB | 45.9 | 47.1 | 79.7 | 75.6 | 68.8 | 77.3 | 42.3 |
| | | | 1-bit | **2.0 GB (10.6×)** | 0.4 kB | 45.7 | 47.0 | 79.3 | 75.6 | 69.1 | 77.6 | 41.1 |
| LLaMA2 -13B | QLoRA | Baseline | 16-bit | 8.2 GB | 0 | **55.0** | **50.3** | 80.0 | **79.2** | 71.5 | 80.9 | **39.1** |
| | | ALAM | 1-bit | **0.8 GB (10.3×)** | 0.5kB | 54.8 | 50.0 | 80.2 | 78.8 | **71.6** | **81.0** | 38.9 |
| LLaMA -30B | QLoRA | Baseline | 16-bit | 15.8 GB | 0 | 56.6 | 52.4 | **81.4** | **82.1** | 75.0 | **82.7** | **42.8** |
| | | ALAM | 1-bit | **1.6 GB (9.9×)** | 0.7kB | **56.7** | **52.7** | 80.8 | 81.8 | **75.3** | 82.5 | 42.5 |

Table 4: Comparisons with model compression techniques in fine-tuning LLMs using a single NVIDIA A6000 GPU. The micro-batch size is set to 8.

| Model | Training Strategy | Average accuracy | Memory usage | | |
|---|---|---|---|---|---|
| | | | Model state | Activation | Total |
| LLaMA2-7B | Full fine-tuning | Out of memory | 52 GB | 21 GB | 73 GB |
| | LoRA | 62.5 | 13 GB | 21 GB | 34 GB |
| | LoRA + ALAM (1-bit) | 62.2 | 13 GB | 2 GB | **15 GB** |
| LLaMA2-13B | Full fine-tuning | Out of memory | 104 GB | 33 GB | 137 GB |
| | QLoRA | 65.1 | 7 GB | 33 GB | 40 GB |
| | QLoRA + ALAM (1-bit) | 65.0 | 7 GB | 3 GB | **10 GB** |

We also compared the training memory usage of the model compression techniques and ALAM in Table 4. The results shows that LoRA and QLoRA only reduce the model state memory which includes parameters, gradients, and optimizer states compared to baseline. ALAM, on the other hand, compresses activation memory, achieving $10.6\times$ and $10.3\times$ compression rates on LLaMA2-7B and 13B, respectively. By applying ALAM to LoRA and QLoRA, we can reduce total memory by 56% and 75%, while maintaining comparable accuracy for LLaMA2-7B and 13B, respectively.

**Usability** To demonstrate the utility of ALAM, we first observed how much our ALAM could increase the micro-batch size in a single NVIDIA RTX-3090 Ti GPU when fine-tuning LLaMA2-7B using LoRA. Fig. 8 shows that ALAM allows for using the micro-batch size of 16, improving the baseline and GACT by $8\times$ and $2\times$, respectively. We also simulated how much memory could be reduced for fixed micro-batch sizes when fine-tuning LLaMA2-7B using LoRA, as shown in Table 5. The size of activation memory increases with the micro-batch size and sequence length, whereas the model state memory storing parameters, gradients, and optimizer state remains unchanged. Consequently, the memory savings of ALAM is more noticeable with larger micro-batch sizes, as demonstrated in Table 5. For instance, when a micro-batch size is 8, ALAM reduces the total memory by 56% and 21% compared to the baseline and GACT, respec-

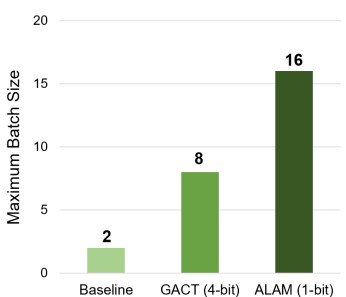

Figure 8: The maximum micro-batch size of different training schemes for fine-tuning LLaMA2-7B.

tively, but the saving increases to 84% and 52% when the micro-batch size is 64.

Table 5: Activation memory and total memory required for fine-tuning LLaMA2-7B using LoRA with different training schemes.

| Scheme | Precision | Micro-batch size = 8 | | Micro-batch size = 32 | | Micro-batch size = 64 | |
|---|---|---|---|---|---|---|---|
| | | Activation | Total | Activation | Total | Activation | Total |
| Baseline | 16-bit | 21 GB | 34 GB | 84 GB | 98 GB | 169 GB | 182 GB |
| GACT | 4-bit | 6 GB | 19 GB | 24 GB | 37 GB | 47 GB | 60 GB |
| ALAM | 1-bit | **2 GB** | **15 GB** | **8 GB** | **21 GB** | **16 GB** | **29 GB** |

## 7 DISCUSSION

In this paper, we introduce ALAM, a new activation-compressed training framework for transformer models. Our ALAM enables deeper compression of activations with low sensitivity through Average Quantization and drastically reduces the memory required for sensitivity calculations using the GradNormVar algorithms. As a result, ALAM reduced the activation memory of LLaMA-30B by $9.9\times$ while outperforming GACT, the state-of-the-art ACT framework, with comparable accuracy. We hope that our ALAM will boost the training of LLMs, overcoming the challenges of limited GPU resources.

**Limitations** ACT frameworks require additional computation to compress and decompress activations. This complexity contributed to a 11-23% increase in training time for both GACT and ALAM (see Appendix C). In our next study, we will address these challenges for efficient training.

REPRODUCIBILITY

In Sections 3 and 4, we elaborated in detail on the core components of ALAM, such as Average Quantization and GradNormVar. Section 5 describes the operation of the ALAM system, while the experimental results in Section 6 can be replicated by referring to the hyperparameters and other experimental settings detailed in Appendix D. Additionally, we provide the reproducible code for ALAM at https://github.com/WooSunghyeon/alam.

ACKNOWLEDGMENT

This work was supported by the National Research Foundation of Korea (Grant NRF-2022R1C1C1006880) and the Institute of Information & communications Technology Planning & Evaluation (Grants 2021-0-01343-004 and IITP-2023-RS-2023-00256081).

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

APPENDICES

## A    PROOF OF THEOREM

In this section, we first establish the relationship between gradient variance and gradient difference in activation compression training (ACT) depicted in Eq. 5. We then derive the boundary function of gradient variance for fully-connected, convolution, batch normalization, and layer normalization layers. We confirm that all of these boundary functions can be formulated as Eq. 6. Finally, we prove Theorem 1.

### A.1    RELATIONSHIP BETWEEN GRADIENT VARIANCE AND GRADIENT DIFFERENCE IN ACT

Chen et al. (2021) proved that the gradient variance of ACT is unbiased as follows:

$$\mathbb{E}\big[\Delta\nabla_\theta\mathcal{L}\big] = \mathbb{E}\big[\hat{\nabla}_\theta\big] - \mathbb{E}\big[\nabla_\theta\big] = 0 \tag{7}$$

Assume two random vectors $X = (x_n) \in \mathbb{R}^N$ and $Y = (y_n) \in \mathbb{R}^N$. If $\mathbb{E}\big[y_n\big] = 0 \ \forall n$, then $\mathbb{E}\big[\|X + Y\|^2\big]$ can be expressed as

$$
\begin{aligned}
\mathbb{E}\big[\|X + Y\|^2\big] &= \mathbb{E}\left[\sum_{n=1}^N (x_n + y_n)^2\right] \\
&= \mathbb{E}\left[\sum_{n=1}^N x_n^2 + 2x_n y_n + y_n^2\right] \\
&= \mathbb{E}\big[\|X\|^2\big] + \mathbb{E}\big[\|Y\|^2\big] + 2\sum_{n=1}^N \mathbb{E}[x_n]\mathbb{E}[y_n] \\
&= \mathbb{E}\big[\|X\|^2\big] + \mathbb{E}\big[\|Y\|^2\big]
\end{aligned}
\tag{8}
$$

By adopting this property, we can derive the gradient variance of ACT as below.

$$
\begin{aligned}
\mathrm{Var}\big[\hat{\nabla}_\theta\mathcal{L}\big] &= \mathbb{E}\big[\|\hat{\nabla}_\theta\mathcal{L}\|^2\big] - \|\mathbb{E}\big[\hat{\nabla}_\theta\mathcal{L}\big]\|^2 \\
&= \mathbb{E}\big[\|\nabla_\theta\mathcal{L} + \Delta\nabla_\theta\mathcal{L}\|\big]^2 - \|\mathbb{E}\big[\hat{\nabla}_\theta\mathcal{L}\big]\|^2 \\
&= \mathbb{E}\big[\|\nabla_\theta\mathcal{L}|^2\big] + \mathbb{E}\big[\|\Delta\nabla_\theta\mathcal{L}|^2\big] - \|\mathbb{E}\big[\hat{\nabla}_\theta\mathcal{L}\big]\|^2 \\
&= \mathbb{E}\big[\|\nabla_\theta\mathcal{L}|^2\big] + \mathbb{E}\big[\|\Delta\nabla_\theta\mathcal{L}|^2\big] - \|\mathbb{E}\big[\nabla_\theta\mathcal{L}\big]\|^2 \\
&= \mathrm{Var}\big[\nabla_\theta\mathcal{L}\big] + \mathbb{E}\big[\|\Delta\nabla_\theta\mathcal{L}|^2\big]
\end{aligned}
\tag{9}
$$

### A.2    BOUNDARY FUNCTION OF GRADIENT VARIANCE

Evans & Aamodt (2021) suggested that compressing activations would increase the gradient variance, and minimizing the gradient variance between actual and approximated activations can lead to convergence of a network trained using approximated activations. Therefore, they derived the boundary function of the gradient variance for widely used layers such as fully-connected, convolution, batch normalization, and layer normalization layers. Here we summarize their findings and extend them to generalize the boundary function as Eq. 6.

**Fully-Connected Layer**    Assuming weight $\theta = (\theta_{h_2 h_1}) \in \mathbb{R}^{H_2 \times H_1}$, input activation $X = (x_{nh_1}) \in \mathbb{R}^{N \times H_1}$, and gradient of output activation $\nabla_Y f = (\partial f / \partial y_{nh_2} \in \mathbb{R}^{N \times H_2})$, Evans & Aamodt (2021) derive the boundary function of $\|\Delta\nabla_\theta f\|^2$ as

$$\|\Delta\nabla_\theta f\|^2 \leq \|\nabla_Y f\|^2 \|\Delta X\|^2 \tag{10}$$

Approximating activations does not modify the gradient of output in fully-connected layers, and hence we treat $\|\Delta\nabla_\theta f\|^2$ as a constant. Therefore, the boundary function can be expressed as $D(\Delta X) = d\|\Delta X\|^2$ where $d$ is a constant.

**Convolution**   Evans & Aamodt (2021) also find the boundary function of a convolutional layer with filter size $R \times S$, stride $T$, input activation $X = (x_{nc_1 hw}) \in \mathbb{R}^{N \times C_1 \times H \times W}$, and the gradient of the output activation $\nabla_Y f = (\partial f / \partial y_{nc_2 hw} \in \mathbb{R}^{N \times C_2 \times H \times W})$.

$$\|\Delta\nabla_\theta f\|^2 \leq \frac{RS}{T}\|\nabla_Y f\|^2\|\Delta X\|^2 \tag{11}$$

The gradient of the output in a convolutional layer can be treated as a constant, similar to that of the fully-connected layer. Hence, we can simplify the boundary function to $D(\Delta X) = d\|\Delta X\|^2$, where $d$ represents a constant.

**Batch Normalization**   While approximating activations ($X \to X + \Delta X$) does not directly impact error propagation (i.e., $\frac{\partial f}{\partial x_i}$ does not have any terms related to $x_i$) in fully-connected and convolutional layers, it does modify error propagation in batch normalization (i.e.. $\frac{\partial f}{\partial x_i}$ has terms related to $x_i$). Therefore, Evans & Aamodt (2021) analyzed the gradient variance of error propagation, where the weight is $(\gamma_c) \in \mathbb{R}^C$, the input activation is $X = (x_{nchw}) \in \mathbb{R}^{N \times C \times H \times W}$, and the gradient of the output activation is $\nabla_Y f = (\partial f / \partial y_{nchw} \in \mathbb{R}^{N \times C \times H \times W})$. The boundary function is

$$\|\Delta\nabla_X f\|^2 \leq \sum_{n,c,h,w} \frac{2s_c^4\gamma_c^2}{M^2}g_c^2\Delta x_{nchw}^2 \tag{12}$$

where $s_c$ is the the inverse of the standard deviation of the input activation across the batch dimension $(\sigma_c^2 + 10^{-5})^{1/2}$, $M$ is $NHW$, and $g_c$ is the upper bound of the gradient variance by approximating activation (i.e., $(\Delta\frac{\partial f}{\partial \gamma_c})^2 + (\frac{\partial f}{\partial \gamma_c})^2 \leq g_c^2$).

We simplify the inequality in Eq. 12 further by utilizing the inequality below:

$$\sum_i a_i b_i \leq \sum_i a_i \sum_i b_i \quad s.t.\forall a_i, \forall b_i \geq 0 \tag{13}$$

By applying the inequality above to Eq. 12, we obtain

$$\begin{aligned}
\|\Delta\nabla_X f\|^2 &\leq \sum_{n,c,h,w} \frac{2s_c^4\gamma_c^2}{M^2}g_c^2\Delta x_{nchw}^2 \\
&\leq \frac{1}{M^2}\sum_{n,c,h,w} 2s_c^4\gamma_c^2 g_c^2 \sum_{n,c,h,w}\Delta x_{nchw}^2 = \frac{1}{M}(\sum_c 2s_c^4\gamma_c^2 g_c^2)\,\|\Delta X\|^2
\end{aligned} \tag{14}$$

In addition, we derive the boundary function of the gradient variance of parameters below.

$$\begin{aligned}
\|\Delta\nabla_\gamma f\|^2 &= \sum_c s_c^2(\sum_{n,h,w}\frac{\partial f}{\partial y_{nchw}}\Delta x_{nchw})^2 \\
&\leq \sum_c s_c^2 \sum_c (\sum_{n,h,w}\frac{\partial f}{\partial y_{nchw}}\Delta x_{nchw})^2 \\
&\leq \sum_c s_c^2 \sum_c \left(\sum_{n,h,w}(\frac{\partial f}{\partial y_{nchw}})^2 \sum_{n,h,w}\Delta x_{nchw}^2\right) \\
&\leq \sum_c s_c^2 \sum_{n,c,h,w}(\frac{\partial f}{\partial y_{nchw}})^2 \sum_{n,c,h,w}\Delta x_{nchw}^2 = (\sum_c s_c^2)\|\nabla_Y f\|^2\|\Delta X\|^2
\end{aligned} \tag{15}$$

We applied the inequality in Eq. 13, followed by the Cauchy–Schwarz inequality, and then applied the inequality in Eq. 13 again to find the boundary function. Finally, we can write the boundary functions of batch normalization in Eqs. 14 and 15 as $d\|\Delta X\|^2$, where $d$ is a constant.

**Layer Normalization**   Modifying activations affects the gradient variance of error propagation in layer normalization as well (Evans & Aamodt, 2021). Assuming weight of layer normalization $(\gamma_c) \in \mathbb{R}^C$, input activation $X = (x_{nc}) \in \mathbb{R}^{N \times C}$, and the gradient of the output activation $\nabla_Y f = (\partial f / \partial y_{nc}) \in \mathbb{R}^{N \times C}$, the boundary function is

$$\|\Delta \nabla_X f\|^2 \simeq \sum_{n,c} \frac{s_n^4 \gamma_c^2 \Delta x_{nc}^2}{C} \tag{16}$$

, where $s_n$ denotes the the inverse of the standard deviation of the input activation across the feature dimension $(\sigma_n^2 + 10^{-5})^{1/2}$. We additionally applied Eq. 13 to Eq. 16 derived by Evans & Aamodt (2021) for simplicity as follows:

$$\begin{aligned}
\|\Delta \nabla_X f\|^2 &\simeq \sum_{n,c} \frac{s_n^4 \gamma_c^2 \Delta x_{nc}^2}{C} \\
&\leq N \sum_n s_n^4 \sum_c \gamma_c^2 \sum_{n,c} x_{nc}^2 = N(\sum_n s_n^4 \sum_c \gamma_c^2) \, \|\Delta X\|^2
\end{aligned} \tag{17}$$

We further derive the gradient variance of parameters, as shown below.

$$\begin{aligned}
\|\Delta \nabla_\gamma f\|^2 &= \sum_c (\sum_n s_n \frac{\partial f}{\partial y_{nc}} \Delta x_{nc})^2 \\
&\leq \sum_c (\sum_n s_n^2 \sum_n (\frac{\partial f}{\partial y_{nc}})^2 \sum_n \Delta x_{nc}^2) \\
&\leq C(\sum_n s_n^2) \, \|\nabla_Y f\|^2 \|\Delta X\|^2
\end{aligned} \tag{18}$$

The Cauchy-Schwarz inequality and Eq. 13 are applied to find the boundary function of gradient variance of parameters in layer normalization. Consequently, the boundary function of layer normalization in Eqs. 17 and 18 can be represented as $d\|\Delta X\|^2$, where $d$ is a constant.

In summary, the boundary function of gradient variance in a general layer can be expressed as Eq. 6. We do not consider layers such as ReLU and MaxPool that only require a 1-bit mask for backward pass, as these layers do not require activation compression.

### A.3   MINIMIZING GRADIENT VARIANCE

Consider activation $X = \{x_i^{(g)} | 1 \leq i \leq N_g, 1 \leq g \leq G\}$ where $G$ and $N_g$ denote the number of groups and the number of elements in group $g$, respectively. The elements in each group are approximated to an arbitrary single value as $X + \Delta X = \{\hat{x}^{(g)}, \hat{x}^{(g)}, \ldots, \hat{x}^{(g)} | 1 \leq g \leq G\}$. In this case, we can prove Theorem 1 using Eq. 6, as detailed below.

**Theorem 1.** *The value of the boundary function of gradient difference is minimized when each group of activations is approximately represented using its group average value.*

*Proof.* We define activation as $X = \{x_i^{(g)} | 1 \leq i \leq N_g, 1 \leq g \leq G\}$ and compressed activation as $X + \Delta X = \{\hat{x}^{(g)}, \hat{x}^{(g)}, \ldots, \hat{x}^{(g)} | 1 \leq g \leq G\}$. Then, Eq. 6 can be expressed as

$$D(\Delta X) = d\|\Delta X\|^2$$
$$= d\sum_{g=1}^{G}\sum_{i=1}^{N_g}(\hat{x}^{(g)} - x_i^{(g)})^2$$

Therefore, $D(\Delta X)$ is a convex function with respect to $\hat{x}^{(g)}$, and we can minimize the value of $D(\Delta X)$ when the condition $\partial D(\Delta X)/\partial \hat{x}^{(g)} = 0$ is satisfied. Then,

$$\frac{\partial D(\Delta X)}{\partial \hat{x}^{(g)}} = d\sum_{g=1}^{G}\sum_{i=1}^{N_g}\frac{\partial}{\partial \hat{x}^{(g)}}(\hat{x}^{(g)} - x_i^{(g)})^2$$
$$= 2d\sum_{g=1}^{G}\sum_{i=1}^{N_g}(\hat{x}^{(g)} - x_i^{(g)})$$
$$= 2d\sum_{g=1}^{G}(N_g\hat{x}^{(g)} - \sum_{i=1}^{N_g}x_i^{(g)}) = 0$$

Consequently, we obtain

$$\hat{x}^{(g)} = \frac{\sum_{i=1}^{N_g}x_i^{(g)}}{N_g} \tag{19}$$

which is the average of each group. In other words, we can minimize the boundary function by approximating the elements in each group to their group average.

□

## B  RESPONSES GENERATED BY LLAMA2-7B FINE-TUNED USING ALAM

**Prompt** : Translate the sentence 'Deux ou trois choses que je sais d'elle' into English.

**Baseline** : I know two or three things about her.

**ALAM (1-bit)** : Two or three things that I know about her.

**Prompt** : Recommend a movie for me to watch during the weekend and explain the reason.

**Baseline** : Movie: Spiderman: No way home
- Reason: If you are a fan of Marvel universe, Spiderman: No way home is the movie that you should watch during the weekend. Spiderman: No way home tells the story of Peter's friend from high school and his classmate that get evicted from their homes, so they steal something from Doctor Strange's transport vehicle, called the Sanctum Sanctorum. Doctor Strange uses the Pretorians to capture the thieves and return the stolen item. But the Pretorians overpower Doctor Strange and take him to the Void, where they torture him. The thieves are brought to the Sanctum Sanctorum and Doctor Strange grants them their wish. For Peter, he puts Doctor Strange's spell on him so that he can retrieve everyone who is stuck in different dimensions.

**ALAM (1-bit)** : There are a few suggestions that I would recommend you to watch this weekend. I think you would enjoy both of them; "Spiderman" and "Avatar". Spiderman is considered to be one of the best Marvel films. It is about a teenager who is bitten by a radioactive spider and gains a few superhuman powers that have helped him save the world from a few villains. Avatar is one of the most successful films of the 21st century. The film is centred around a human who infiltrates the world of the Na'vi, the indigenous inhabitants of a planet known as Pandora in the twenty-first century on Earth.

## C  TRAINING TIME OF ACTIVATION-COMPRESSED TRAINING

In this section, we analyze the time overhead of activation compressed training including ALAM. Note that the sensitivity calculation is conducted only twice (before training and at 10% of the training process), and hence its overhead is excluded from the analysis. Our ALAM utilizes average quantization, which involves average calculation and quantization including searching for min and max values, multiplication with a scaling factor, and stochastic rounding. These operations have a time complexity of $\mathcal{O}(n)$, where $n$ represents the number of activations. Korthikanti et al. (2022) calculated the FLOPs and the number of activations of a transformer block with flash-attention Dao et al. (2022) as $24bsh^2 + 4bs^2h$ and $34bsh$, respectively, where $b$, $s$, and $h$ denote the micro-batch size, sequence length, and hidden size, respectively. Based on this and the fact that the size of the hidden dimension is generally larger than the sequence length in LLMs, the time complexity of training an LLM and the additional time complexity due to ALAM are calculated as shown in Table 6 below.

Table 6: Time complexity of training and activation compression required by ALAM.

|  | Training | Activation compression |
|---|---|---|
| Time complexity | $\mathcal{O}(bsh^2)$ | $\mathcal{O}(bsh)$ |

Therefore, as the model size $h$ increases, the training time overhead due to ALAM becomes relatively smaller. To verify this, we trained models larger than 7B on the LIMA (1K) dataset for 5 epochs, and the results are displayed in Table E.2 below. Table 7 shows that while there is a 23% time overhead for the LLaMA2-7B model, the overhead is reduced to 18% and 12% for the LLaMA2-13B and LLaMA-30B models, respectively.

Table 7: Time complexity of training and activation compression required by ALAM.

| Model | Baseline | GACT (4-bit) | ALAM (1-bit) |
|---|---|---|---|
| LLaMA2-7B | 30 min | 36 min | 37 min |
| LLaMA2-13B | 51 min | 58 min | 60 min |
| LLaMA-30B | 115 min | 127 min | 128 min |

# D EXPERIMENTAL DETAILS

**Training MLP to confirm Theorem 1**   We trained an MLP (Haykin, 1994) 512-512-10 model on the MNIST dataset (Deng, 2012) using various approximate activation methods such as averaging and median. In this experiment, training was executed with a fixed learning rate for 30 epochs using the SGD optimizer (Ruder, 2016). The batch size and learning rate were set to 128 and 0.01, respectively. A single RTX-3090 Ti GPU was used.

**Training VGG-11 to confirm quality of activations**   We conducted experiments to verify the quality of activations compressed by Average Quantization in VGG-11 (Simonyan & Zisserman, 2015) when trained on the CIFAR-100 dataset (Krizhevsky et al., 2009). The learning rate and batch size were set at 0.01 and 128, respectively, and the SGD optimizer (Ruder, 2016) was utilized. The training lasted for 100 epochs, and we scheduled the learning rate using cosine annealing (Loshchilov & Hutter, 2016). This experiment was conducted using a single RTX-3090 Ti GPU.

**Fine-tuning BERT models on the datasets in the GLUE benchmark**   We evaluated the performance of ALAM by fine-tuning various BERT models on the datasets in the GLUE benchmark (Wang et al., 2019). We trained DistilBERT-Base (Sanh et al., 2019) on the STS-B and WNLI datasets, using the distilbert-base-uncased as the pretrained model from HuggingFace (Wolf et al., 2020). We set the batch size to 16 and trained for 5 epochs with different learning rates ($5 \times 10^{-5}$, $1 \times 10^{-4}$), and we reported the highest accuracy. Furthermore, we trained the BERT-Base model (Devlin et al., 2019) on the CoLA, RTE, and MRPC datasets, utilizing the bert-base-cased from HuggingFace (Wolf et al., 2020) as our pretrained model. With a batch size of 16, we trained for 3 epochs varying learning rates ($2 \times 10^{-5}, 3 \times 10^{-5}, 4 \times 10^{-5}$, and $5 \times 10^{-5}$) and reported the highest accuracy. The BERT-Large model (Devlin et al., 2019) was trained on the SST-2, QNLI, MNLI, and QQP datasets using the bert-base-cased as the pretrained model from HuggingFace (Wolf et al., 2020). We trained with a batch size of 16 for 10 epochs at a learning rate of $1 \times 10^{-5}$, a setting proposed by Mosbach et al. (2021) for robust training. All experiments were conducted on a single RTX-3090 Ti GPU, and we reported the results as the average across 5 seeds.

**Fine-tuning large language models on the Alpaca dataset**   To demonstrate the effectiveness of ALAM in training large language models, we conducted the following experiments: We fine-tuned LLaMA2-7B model (Touvron et al., 2023b) on the Alpaca dataset (Taori et al., 2023) for 1 epoch with a learning rate of $1 \times 10^{-4}$. Instead of the full fine-tuning, we used LoRA (Hu et al., 2022), which is a parameter-efficient fine-tuning (PEFT) technique. Similarly, we fine-tuned the LLaMA2-13B model (Touvron et al., 2023b) and LLaMA-30B model (Touvron et al., 2023a) on the LIMA dataset (Zhou et al., 2023) for 5 epochs with a learning rate of $5 \times 10^{-5}$ and $2 \times 10^{-5}$, respectively. In this experiment, we applied the QLoRA (Dettmers et al., 2023) as PEFT.

All experiments were carried out with a max sequence length of 512, utilizing a single NVIDIA A6000 GPU. One exception is the experiments in Section 6, where we specifically investigated how much we could increase the micro-batch size using a single NVIDIA RTX-3090 Ti GPU. The fine-tuned models were then evaluated to obtain 5-shot accuracy for MMLU (Hendrycks et al., 2021) and 0-shot accuracy for common sense reasoning (Bhakthavatsalam et al., 2021; Bisk et al., 2020; Zellers et al., 2019; Win, 2019), reading comprehension (Clark et al., 2019), and truthfulness tasks (Lin et al., 2021) using the lm-harness library[1].

---

[1]https://github.com/EleutherAI/lm-evaluation-harness

