# OpenReview forum: "ALAM: Averaged Low-Precision Activation for Memory-Efficient Training of Transformer Models"
_ICLR.cc/2024/Conference — ICLR 2024 poster_

### Official Review · Reviewer_7RJR · 2023-10-31

**Soundness:** 3 good
**Presentation:** 3 good
**Contribution:** 3 good
**Rating:** 6
**Confidence:** 4

**Summary:**

This paper proposes a novel ACT framework--ALAM to quantize activations for memory-efficient training of transformer models. ALAM utilizes average quantization and a lightweight sensitivity calculation scheme. The experiments show that ALAM significantly reduces activation memory.

**Strengths:**

1.	The proposed ALAM provides high-quality deeply compressed activations with precision of less than 1 bit, which enables further compression without sacrificing accuracy.
2.	The proposed ALAM is demonstrated to successfully compress activations in various transformer models.

**Weaknesses:**

The complexity of ALAM is not analyzed. And the paper only compares training time for LLaMA-3B.

**Questions:**

Please refer to the weakness.

---

> ### Author Response · Authors · 2023-11-18
> **Response to Reviewer 7RJR**
>
> **We thank the reviewer for carefully reviewing our submission and providing valuable feedback. Please see below for our response to the questions and comments.**
>
> **Q1)** The complexity of ALAM is not analyzed. And the paper only compares training time for LLaMA-3B.
>
> **A1)** We thank the reviewer for this valuable feedback. To address the reviewer’s concern, here we first analyze the time complexity of ALAM in more detail. Please note that the sensitivity calculation is conducted only twice (before training and at 10% of the training process), and hence its overhead is excluded from the analysis. Our ALAM utilizes average quantization, which involves average calculation and quantization including searching for min and max values, multiplication with a scaling factor, and stochastic rounding. These operations have a time complexity of $\mathcal{O(n)}$, where n represents the number of activations. The authors in [1] calculated the FLOPs and the number of activations of a transformer block with flash-attention as $24bsh^{2} + 4bs^{2}h$ and $34bsh$, respectively, where $b$, $s$, and $h$ denote the batch size, sequence length, and hidden size, respectively. Based on this and the fact that the size of the hidden dimension is generally larger than the sequence length in LLMs, the time complexity of training an LLM and the additional time complexity due to ALAM are calculated as shown in Table E.1 below.
>
> **Table E.1. Time complexity of training and activation compression required by ALAM.**
>
> |                 | Training   | Activation compression |
> |-----------------|------------|------------------------|
> | Time complexity | $\mathcal{O}(bsh^{2})$ | $\mathcal{O}(bsh)$                 |
>
> Therefore, as the model size $h$ increases, the training time overhead due to ALAM becomes relatively smaller. To verify this, we trained models larger than 7B on the LIMA (1K) dataset for 5 epochs, and the results are displayed in Table E.2 below.
>
> **Table E.2. Time for fine-tuning LLaMA2-7B for 5 epochs on the LIMA datasets using QLoRA with different ACT frameworks with a single NVIDIA A6000 GPU. The micro batch size is set to 2.**
>
> | Model | Baseline | GACT (4-bit) | ALAM (1-bit) |
> |---|---|---|---|
> | LLaMA2-7B | 30 min | 36 min | 37 min |
> | LLaMA2-13B | 51 min | 58 min | 60 min |
> | LLaMA-30B | 115 min | 127 min | 128 min |
>
> Table E.2 shows that while there is a 23% time overhead for the LLaMA2-7B model, the overhead is reduced to 18% and 12% for the LLaMA2-13B and LLaMA-30B models, respectively. We once again thank the reviewer for this great suggestion, and we will include this analysis in the camera-ready version if the paper is accepted.
>
> [1] Korthikanti, V. A., Casper, J., Lym, S., McAfee, L., Andersch, M., Shoeybi, M., & Catanzaro, B. Reducing activation recomputation in large transformer models. *Proceedings of Machine Learning and Systems*, *5, 2023*.
>
> [ref1]: Aspose.Words.e37027c5-5038-4eae-ad68-a6473d90ddd2.002.png
> [ref2]: Aspose.Words.e37027c5-5038-4eae-ad68-a6473d90ddd2.006.png

---

> ### Author Response · Authors · 2023-11-23
>
> Dear reviewer,
>
> Thank you once again for carefully reviewing our submission. As the reviewer-author discussion period ends in less than 12 hours, we would be very grateful if we could hear back for any thoughts. If possible, please take a look at our responses and see if they adequately address your concerns. We are also open to any further feedback or queries you may have.
>
> Thank you,
>
> Authors of Paper #3660

---

### Official Review · Reviewer_UZqy · 2023-10-31

**Soundness:** 3 good
**Presentation:** 3 good
**Contribution:** 2 fair
**Rating:** 6
**Confidence:** 5

**Summary:**

This paper proposes a method for reducing the memory usage of activations during the training of large-scale models. The techniques extend well-established methods like GACT, further improving the activation compression rate by introducing group-wise quantization. The methods are evaluated on popular transformer models such as BERT and LLaMa, demonstrating enhanced compression rate while preserving baseline accuracy.

**Strengths:**

1). The paper is well-written.

2). The compression rates have been improved compared with previous methods.

**Weaknesses:**

The primary concern is the lack of novelty. The method relies heavily on existing frameworks, such as GACT and ActNN, using the same concept of quantization for activation compression. Additionally, the proposed quantization method is also a well-known clustering approach. The theoretical analysis, suggesting the minimization of quantization MSE (as shown in quation 6) appears rather straightforward. While the authors did introduce some tricks, such as gradient normalization variance, to improve the existing framework, the overall technical contribution seems to be relatively modest.

**Questions:**

Could the authors explain the overhead involved in doing the group-wise quantization?

---

> ### Author Response · Authors · 2023-11-18
> **Response to Reviewer UZqy**
>
> **We thank the reviewer for carefully reviewing our submission and providing valuable feedback. Please see below for our response to the questions and comments.**
>
> **Q1)** The primary concern is the lack of novelty. The method relies heavily on existing frameworks, such as GACT and ActNN, using the same concept of quantization for activation compression. Additionally, the proposed quantization method is also a well-known clustering approach. The theoretical analysis, suggesting the minimization of quantization MSE (as shown in equation 6) appears rather straightforward. While the authors did introduce some tricks, such as gradient normalization variance, to improve the existing framework, the overall technical contribution seems to be relatively modest.
>
> **A1)** We thank the reviewer for this valuable feedback. We agree with the reviewers that our AQ algorithm has been developed and proven to work in a similar way to ActNN [1] and GACT [2]. However, we would like to point out that **the core idea of our algorithm, simply compressing a group of adjacent activations using a group average value without the need for extensive iterations to categorize similar values, is entirely new and has not been reported in the literature**. In other words, to the best of our knowledge, our work is the first attempt to compress activations through simple averaging.
>
> Unlike a previous method [3] that employs K-means clustering to group weights, which requires a time-consuming process to find weights with similar values, our novel activation compression method relies solely on simple averaging of adjacent activations without the need to identify similar values, which can be very efficiently implemented. In addition, this approach is validated mathematically, proving that simple averaging can minimize gradient variances.
>
> We also agree that the minimization of quantization MSE in Eq. 6 looks straightforward, but actually deriving it was not very straightforward, as shown in Appendix A.2. More specifically, we mathematically proved that the gradient variance of batch normalization and layer normalization has the form of Eq. 6, which was not proven in prior work [1, 2, 4]. This was a key contribution that allowed us to prove that representing activation as a group average can minimize gradient variance for the first time.
>
> Lastly, to the best of our knowledge, **we are the first to demonstrate an effective activation compression algorithm that supports LLMs with more than 7B parameters**. Unlike model compression algorithms like LoRA and QLoRA, previous activation compression algorithms such as ActNN, MESA, and GACT were not proven to work with LLMs. Our experimental results confirm that ALAM can train large language models such as LLaMA2-7B, LLaMA2-13B, and LLaMA-30B even at 1-bit, whereas conventional methods such as GACT fail to train the models.
>
> **Q2)** Could the authors explain the overhead involved in doing the group-wise quantization?
>
> **A2)** This is an important point, and we thank the reviewer for bringing up this issue. Average Quantization generates sub-1b values by first compressing adjacent activations to a group average value and then performing conventional quantization to the group average value. We suspect that the reviewer is wondering if there is any additional memory or time overhead involved in this process. First, there is no additional memory overhead for compressing activation. As explained in Sections 3.1 and 3.2, we group adjacent activations in the flattened activation vector during the forward pass. During the backward pass, they are restored simply by repeating the group average values. For example, when the group size is 2,
>
> - Averaging in the forward pass: [4, 6, 7, 1, 6, 8] -> [5, 4, 7]
> - Restoring in the backward pass: [5, 4, 7] -> [5, 5, 4, 4, 7, 7]
>
> Therefore, there is no need to indicate specific locations, and hence our algorithm does not save any masks. Second, our Average Quantization requires additional time to compute the group average value. However, its time overhead is similar to that of the activation quantization process in conventional activation compression schemes, which is in line with experimental results shown in Table D.1 below.
>
> **Table D.1. Time for fine-tuning LLaMA2-7B over 5 epochs on the LIMA datasets using QLoRA with different ACT frameworks with a single NVIDIA A6000 GPU. The micro batch size is set to 2.**
>
> | Model | Baseline | GACT (4-bit) | ALAM (1-bit) |
> |---|---|---|---|
> | LLaMA2-7B | 30 min | 36 min | 37 min |
> | LLaMA2-13B | 51 min | 58 min | 60 min |
> | LLaMA-30B | 115 min | 127 min | 128 min |

---

> > ### Author Response · Authors · 2023-11-18
> > **Response to Reviewer UZqy**
> >
> > [1] Jianfei Chen, Lianmin Zheng, Zhewei Yao, Dequan Wang, Ion Stoica, Michael W. Mahoney, & Joseph Gonzalez. ActNN: Reducing Training Memory Footprint via 2-Bit Activation Compressed Training. In ICML 2021, Virtual Event (pp. 1803–1813). PMLR.
> >
> > [2] Xiaoxuan Liu, Lianmin Zheng, Dequan Wang, Yukuo Cen, Weize Chen, Xu Han, Jianfei Chen, Zhiyuan Liu, Jie Tang, Joey Gonzalez, Michael W. Mahoney, & Alvin Cheung. GACT: Activation Compressed Training for Generic Network Architectures. In  ICML 2022, Baltimore, Maryland, USA (pp. 14139–14152). PMLR.
> >
> > [3] Fabien Cardinaux, Stefan Uhlich, Kazuki Yoshiyama, Javier Alonso García, Lukas Mauch, Stephen Tiedemann, Thomas Kemp, & Akira Nakamura. Iteratively Training Look-Up Tables for Network Quantization. IEEE J. Sel. Top. Signal Process. 2020, 14(4), 860–870.
> >
> > [4] R. David Evans, & Tor M. Aamodt. AC-GC: Lossy Activation Compression with Guaranteed Convergence. In NeurIPS 2021, virtual (pp. 27434–27448).

---

> ### Author Response · Authors · 2023-11-23
>
> Dear reviewer,
>
> Thank you once again for carefully reviewing our submission. As the reviewer-author discussion period ends in less than 12 hours, we would be very grateful if we could hear back for any thoughts. If possible, please take a look at our responses and see if they adequately address your concerns. We are also open to any further feedback or queries you may have.
>
> Thank you,
>
> Authors of Paper #3660

---

> > ### Comment · Reviewer_UZqy · 2023-11-23
> >
> > Thanks you for the detailed response. Some of my concerns are addressed. I will raise my rating to 6.

---

> > > ### Author Response · Authors · 2023-11-23
> > >
> > > Thank you very much for the response!

---

### Official Review · Reviewer_Eyz9 · 2023-10-31

**Soundness:** 3 good
**Presentation:** 3 good
**Contribution:** 2 fair
**Rating:** 6
**Confidence:** 2

**Summary:**

This paper introduces ALAM, an ACT framework that applies average quantization and a novel sensitivity calculation method to reduce memory usage while preserving the performance of LLMs. By compressing activations to their group averages, ALAM minimizes the impact on gradient variance, enabling deep compression with an effective precision of less than 1 bit. ALAM’s sensitivity calculation is based on the L2 norm of parameter gradients, which is a memory-efficient approach. The paper's experiments show that ALAM can achieve up to 12.5× activation memory compression in LLMs without sacrificing accuracy. This method represents a significant advancement in making the training of large neural networks more memory-efficient.

**Strengths:**

1. The quantization method based on group average value is very easy to understand and very practical, given the experimental demonstration and equation proof in the paper.
2. The training overhead is acceptable, since the efficient sensitivity calculation method is proposed, and the training time is comparable with the baseline.
3. The evaluation result is very promising and the improvement compared with GACT (SOTA work based on activation compression) is very significant.

**Weaknesses:**

1. The evaluation part is not sufficient. This paper only compares the result with GACT, which is based on activation compression. However, for model compression, other techniques like pruning and low-rank compression are widely used.
2. How to choose the parameter group is unclear. Is this parameter related to the sensitivity? What if the adjacent activations have little similarity?

**Questions:**

1. It would be better to provide the evaluation result with other compression techniques.
2. In Table 1, there is an accuracy drop from 98.72 to 96.09 in a small model. However, in the evaluation result, the compressed model retains the accuracy performance. Can you identify the reason for this? This is because the large model has more redundancy or other reasons?

---

> ### Author Response · Authors · 2023-11-18
> **Response to Reviewer Eyz9**
>
> **We thank the reviewer for carefully reviewing our submission and providing valuable feedback. Please see below for our response to the questions and comments.**
>
> **Q1)** The evaluation part is not sufficient. This paper only compares the result with GACT, which is based on activation compression. However, for model compression, other techniques like pruning and low-rank compression are widely used.
>
> **A1)** We would like to thank the reviewer for bringing up an important point. Following the reviewer’s advice, we conducted detailed comparisons between ALAM and other model compression methods. For comparisons, we selected LoRA [1], which is the state-of-the-art low-rank compression method. We also implemented QLoRA [2], which performs 4-bit quantization for weights and only computes weight gradient for LoRA parameters. QLoRA allows for a 4× reduction in the weights with comparable accuracy, making it widely used for fine-tuning LLMs in constrained environments. The results are presented in Table C.1. Please note that ‘average accuracy’ represents the average accuracy over all tasks in Tables A.1 and A.2 in the global response. Also, ‘param mem’ represents the parameter memory occupied by weights, gradients, and optimizer states during training.
>
> **Table C.1. Comparisons with model compression techniques in fine-tuning LLaMA2-7B and LLaMA2-13B on the LIMA dataset. The micro batch size is set to 8.**
>
> | Model      | Strategy       | Average accuracy | Param mem | Act mem | Total mem |
> |------------|----------------|------------------|-----------|---------|-----------|
> | LLaMA2-7B  | Baseline       | OOM              |   52 GB   |  21 GB  |   73 GB   |
> |            | + LoRA         | 62.5             |   13 GB   |  21 GB  |   34 GB   |
> |            | + LoRA + ALAM | 62.2             |   13 GB   |  2.0 GB | **15 GB** |
> | LLaMA2-13B | Baseline       | OOM              |   104 GB  |  33 GB  |   137 GB  |
> |            | + QLoRA        | 65.1             |   6.6 GB  |  33 GB  |   40 GB   |
> |            | + QLoRA + ALAM | 65.0             |   6.6 GB  |  3.2 GB | **10 GB** |
>
> In experiments, LoRA noticeably reduces the memory allocated for gradients and optimizer states, which results in a 4× compression of parameter memory during training. Additionally, QLoRA exhibits a large compression rate of 16× by utilizing 4-bit weights. However, neither LoRA nor QLoRA compresses activation memory. Our ALAM, on the other hand, focuses on compressing activation memory, and hence achieves 10.6× and 10.3× compression rates on LLaMA2-7B and LLaMA2-13B, respectively. As a result, compared to when only LoRA and QLoRA were applied, additionally employing ALAM further saves 56% and 75% of the total memory with comparable accuracy for LLaMA2-7B and LLaMA2-13B, respectively. In summary, our ALAM is orthogonal to other model compression techniques, and they can be applied simultaneously for maximum memory savings.
>
> Please note that only a handful of pruning schemes supporting very large models have been reported in the literature, with limited memory savings. For instance, LLM-Pruner [3] exhibits a significant drop in accuracy even when pruning just 20% of the parameters in LLaMA-7B, translating to similar saving only in the parameter memory.

---

> ### Author Response · Authors · 2023-11-18
> **Response to Reviewer Eyz9**
>
> **Q2)** How to choose the parameter group is unclear. Is this parameter ‘group’ related to the sensitivity? What if the adjacent activations have little similarity?
>
> **A2)** We thank the reviewer for raising an important issue, and we apologize for the confusion. As described in Sections 3.1 and 3.2, we group adjacent activations together after simply flattening the activations, without any other complicated processes. As the reviewer insightfully pointed out, varying the group size (g) and the precision of the group average (n) could produce different bit precisions (AQ-n/g bit). However, in all experiments using Average Quantization, we fixed the group size to 4 and performed 2-bit quantization on the group average value to obtain AQ 0.5-bit, which is an optimal configuration identified in experiments, as shown in Table C.2.
>
> **Table C.2. Performan comparisons of different AQ 0.5-bit configurations when training VGG-11 on CIFAR-100 (n: precision of group average, g: group size).**
>
> | Precision        | FP32     | 1-bit |  |          |    AQ 0.5-bit    |         |          |
> |------------------|----------|-------|------------|----------|--------|---------|----------|
> | AQ scheme [n,g]  | -        | -     | (1, 2)     | (2, 4)   | (4, 8) | (8, 16) | (32, 64) |
> | Accuracy (%)     | **67.7** | 1.0   | 1.0        | **52.3** | 1.87   | 8.07    | 3.08     |
>
> The group size is not related to sensitivity. More specifically, the sensitivity of each layer is first calculated using the GradNormVar algorithm, which determines the layer’s bit precision. In the layers with low sensitivity, we quantize activations into AQ 0.5-bit, where average quantization is performed with a fixed group size of 4 as discussed above.
>
> The reviewer is certainly correct that adjacent activations may show little similarity. This could be mitigated by only grouping activations with similar distributions, but we did not implement this in our algorithm since this process could be time-consuming and require additional memory to save the mask for recovery. Our simple grouping scheme may lead to some loss of information in activations, but the impact on actual accuracy is minimal because ALAM assigns AQ 0.5-bit only to layers with very low sensitivity. Furthermore, the quality of AQ 0.5-bit activations is still higher than that of 1-bit quantized activations, as shown in Table C.2.

---

> > ### Author Response · Authors · 2023-11-18
> > **Response to Reviewer Eyz9**
> >
> > **Q3)** In Table 1, there is an accuracy drop from 98.72 to 96.09 in a small model. However, in the evaluation result, the compressed model retains the accuracy performance. Can you identify the reason for this? This is because the large model has more redundancy or other reasons?
> >
> > **A3)** We apologize for the confusion. Please note that the experiments in Table 1 did not fully implement our ALAM algorithm; Table 1 simply compares the quality of activations obtained by different compressing methods when the group size is 32 and the group average is not quantized. Also, we assigned identical precision to all layers in this experiment, whereas ALAM assigns different precision to the layers based on their sensitivity. This caused a relatively large accuracy drop in Table 1.
> >
> > To fully address the reviewer's concern, we conducted additional experiments of training the same MLP model on MNIST using ALAM, and the experimental results are displayed in Table C.3 below. The results confirm that ALAM successfully trains the MLP model with comparable accuracy to the baseline (FP32). We sincerely appreciate the reviewer’s comments, which have helped us identify unclear aspects of our paper. We will clarify these points in the camera-ready version if the paper is accepted.
> >
> > **Table C.3. Test accuracy of MLP trained on MNIST with different activation approximation methods.**
> > |  | FP | Simple group averaging (g=32) | ALAM |
> > |---|---|---|---|
> > | Accuracy (%) | 98.72 | 96.09 | 98.60 |
> >
> > [1] Edward J. Hu, Yelong Shen, Phillip Wallis, Zeyuan Allen-Zhu, Yuanzhi Li, Shean Wang, Lu Wang, & Weizhu Chen. LoRA: Low-Rank Adaptation of Large Language Models. In *ICLR 2022, Virtual Event, April 25-29, 2022*. OpenReview.net.
> >
> > [2] Tim Dettmers, Artidoro Pagnoni, Ari Holtzman, & Luke Zettlemoyer. QLoRA: Efficient Finetuning of Quantized LLMs*. CoRR, 2023. abs/2305.14314.*
> >
> > [3] Xinyin Ma, Gongfan Fang, & Xinchao Wang. LLM-Pruner: On the Structural Pruning of Large Language Models*. CoRR, 2023. abs/2305.11627.*

---

> ### Author Response · Authors · 2023-11-23
>
> Dear reviewer,
>
> Thank you once again for carefully reviewing our submission. As the reviewer-author discussion period ends in less than 12 hours, we would be very grateful if we could hear back for any thoughts. If possible, please take a look at our responses and see if they adequately address your concerns. We are also open to any further feedback or queries you may have.
>
> Thank you,
>
> Authors of Paper #3660

---

### Official Review · Reviewer_HytY · 2023-11-07

**Soundness:** 3 good
**Presentation:** 3 good
**Contribution:** 3 good
**Rating:** 6
**Confidence:** 2

**Summary:**

This work focuses on the challenge of high GPU memory consumption for storing activation for deep neural network training, especially for large language models (LLMs). Existing activation-compressed training (ACT) approaches introduce significant performance drops when applied to LLMs. This paper proposes a new ACT framework, ALAM, that applies average quantization and a lightweight sensitivity calculation scheme to enable substantial memory savings for LLM training while maintaining training performance. The experiments show that the ALAM framework can support up to a 12.5x compression rate with 1-bit quantization without compromising accuracy.

**Strengths:**

+ The paper is well-written and easy to follow.
+ The proposed average quantization in ACT is simple but effective.
+ The proposed GradNormVar algorithm significantly reduces the memory requirement for storing the gradients during the sensitivity evaluation.
+ The evaluation results cover both the classical model BERT and the latest model LLaMa in terms of large language model training/finetuning. The results of fine-tuning LLaMa-2 with 2-bit and even 1-bit ALAM are promising.

**Weaknesses:**

- It would be better to add QLoRA as a baseline for evaluation.
- It is unclear how the proposed ALAM performs on larger models, such as LLaMa-13B, LLaMa-30B, etc. It would be better to show the results on these larger models.

**Questions:**

Please answer the questions in weaknesses section.

---

> ### Author Response · Authors · 2023-11-18
> **Response to Reviewer HytY**
>
> **We thank the reviewer for carefully reviewing our submission and providing valuable feedback. Please see below for our response to the questions and comments.**
>
> **Q1)** It would be better to add QLoRA as a baseline for evaluation. Also, it is unclear how the proposed ALAM performs on larger models, such as LLaMa-13B, LLaMa-30B, etc.
>
> **A1)** We thank the reviewer for this great suggestion. Following the reviewer’s comment, we applied QLoRA [1] and evaluated our algorithm on larger models: LLaMA2-13B and LLaMA-30B. Please note that we used the recently released LLaMA2 for a 13B model, which has the same structure as LLaMA but is pre-trained on a broader range of datasets, leading to improved accuracy. On the other hand, LLaMA2-30B is not publicly available, and hence LLaMA-30B was employed instead. In this experiment, we fine-tuned the model for 5 epochs on the LIMA dataset. The results are displayed in Table B.1. In the table, 'precision' refers to the target average precision of activation during training as detailed in the manuscript, not the precision of the weight.
>
> **Table B.1. Test accuracy, activation memory (ACT mem) with compression, and memory for calculating sensitivity (sens mem) in fine-tuning LLaMA2-13B and LLaMA-30B on the LIMA dataset. The micro batch size is set to 2.**
>
> | Model       | PEFT  | Scheme   | Precision | Act mem            | MMLU     | Arc-c    | PIQA     | Hellaswag | WinoGrande | BoolQ    | TruthfulQA |
> |-------------|-------|----------|-----------|--------------------|----------|----------|----------|-----------|------------|----------|------------|
> | LLaMA2-13B  | QLoRA | Baesline | 16-bit    | 8.2 GB             | **55.0** | **50.3** |   80.0   |  **79.2** |    71.5    |   80.9   |  **39.1**  |
> |             |       | ALAM     | 1-bit     | **0.8 GB (10.3x)** | 54.8     |   50.0   | **80.2** |    78.8   |  **71.6**  |  **81**  |    38.9    |
> | LLaMA-30B   | QLoRA | Baesline | 16-bit    | 15.8 GB            | 56.6     |   52.4   | **81.4** |  **82.1** |    75.0    | **82.7** |  **42.8**  |
> |             |       | ALAM     | 1-bit     | **1.6 GB (9.9x)**  | **56.7** | **52.7** |   80.8   |    81.8   |  **75.3**  |   82.5   |    42.5    |
>
> Experimental results demonstrate that our ALAM (1-bit) compresses activation by 10.3× in LLaMA2-13B and 9.9× in LLaMA-30B while closely matching the baseline in accuracy for all tasks. We will include these updated experimental results in the camera-ready version if the paper is accepted.
>
> [1] Tim Dettmers, Artidoro Pagnoni, Ari Holtzman, & Luke Zettlemoyer (2023). QLoRA: Efficient Finetuning of Quantized LLMs. CoRR, abs/2305.14314.

---

> ### Author Response · Authors · 2023-11-23
>
> Dear reviewer,
>
> Thank you once again for carefully reviewing our submission. As the reviewer-author discussion period ends in less than 12 hours, we would be very grateful if we could hear back for any thoughts. If possible, please take a look at our responses and see if they adequately address your concerns. We are also open to any further feedback or queries you may have.
>
> Thank you,
>
> Authors of Paper #3660

---

### Author Response · Authors · 2023-11-18
**Response to common questions with additional experimental results (global response)**

We sincerely thank the reviewers for carefully reviewing our submission. Before answering each question in detail, we would like to introduce improved experimental results, which demonstrate the effectiveness of our algorithm on larger models.

**(1) Fine-tuning all layers of LLaMA2-7B (only half of the layers were fine-tuned in the original submission)**

In the original submission, we stated that it was not possible to fine-tune all layers in LLaMA2-7B, and only half of the layers were fine-tuned for this model. **However, we have resolved this issue by slightly optimizing the initialization procedure of our sensitivity calculation algorithm.** Since activations should be temporarily stored during sensitivity calculation, we set the effective precision of all layers to AQ 0.5-bit in the previous experiments to reduce memory overheads. This was acceptable for relatively smaller models as the training still converged at AQ 0.5-bit, but it resulted in training divergence in larger models, rendering the L2 norm of parameter gradients into NaN and producing incorrect sensitivity values. We now avoid this issue by setting the precision of all layers to 2-bit for sensitivity calculation. Since sensitivity calculation is performed with a batch size of 1, the memory required for sensitivity calculation is still smaller than the memory used during actual training with a large batch, and hence the total memory allocated in the GPU does not change due to this modification. Moreover, by solving the divergence problem of the parameter gradients, **we now accurately calculate sensitivity and succeed in fine-tuning all layers of LLaMA2-7B, as shown in Table A.1.** In the updated experiments, we set the maximum sequence length to 512, applied cosine annealing, used LoRA as the baseline parameter efficient fine-tuning (PEFT), and evaluated 5-shot accuracy for MMLU and 0-shot accuracy for common sense reasoning, reading comprehension, and truthfulness tasks.

**Table A.1. Test accuracy, activation memory (ACT mem) with compression, and memory for calculating sensitivity (sens mem) in fine-tuning LLaMA2-7B on the Alpaca dataset. The micro batch size is set to 8.**
| Model      | PEFT | Scheme   | Precision | Act mem            | Sens mem | MMLU     | Arc-c    | PIQA     | Hellaswag | WinoGrande | BoolQ    | TruthfulQA |
|------------|------|----------|-----------|--------------------|----------|----------|----------|----------|-----------|------------|----------|------------|
| LLaMA2-7B  | LoRA | Baesline | 16-bit    | 21.1 GB            |          |   46.0   |   47.6   |   79.5   |  **75.7** |    68.8    | **77.9** |    41.8    |
|            |      | GACT     | 4-bit     | 5.9 GB (3.6x)      | 24 MB    |   44.8   |   46.6   |   78.9   |    75.5   |    68.2    |   77.1   |    40.4    |
|            |      |          | 3-bit     | 4.8 GB (4.4x)      | 24 MB    | fail     | fail     | fail     | fail      | fail       | fail     | fail       |
|            |      | ALAM     | 4-bit     | 5.9 GB (3.6x)      | 0.4 kB   | **46.2** | **47.9** |   79.5   |    75.6   |    69.3    |   77.5   |    42.1    |
|            |      |          | 3-bit     | 4.8 GB (4.4x)      | 0.4 kB   |   45.2   |   47.6   | **79.8** |    75.4   |  **69.5**  |   77.24  |  **42.4**  |
|            |      |          | 2-bit     | 3.4 GB (6.2x)      | 0.4 kB   |   45.9   |   47.1   |   79.7   |    75.6   |    68.8    |   77.3   |    42.3    |
|            |      |          | 1-bit     | **2.0 GB (10.6x)** | 0.4 kB   |   45.7   |   47.0   |   79.3   |    75.6   |    69.1    |   77.6   |    41.1    |

The results above confirm that our ALAM achieves comparable accuracy while compressing the activations to 1-bit, resulting in a 10.6× compression rate in LLaMA2-7B, whereas GACT fails to train the model at 3-bit.

---

> ### Author Response · Authors · 2023-11-18
> **Response to common questions with additional experimental results (global response)**
>
> **(2) ALAM with QLoRA for fine-tuning larger models such as LLaMA2-13B and LLaMA-30B**
>
> Following the comments of the reviewers HytY and Eyz9, we applied the widely-used model compression method, QLoRA [1], and performed experiments on LLaMA2-13B and LLaMA-30B models.  Please note that since the LLaMA2-30B model is not publicly available, we conducted experiments on the LLaMA-30B model instead. In this experiment, we fine-tuned the models for 5 epochs on the LIMA dataset. The experimental results are displayed in Table A.2, where 'precision' refers to the target average precision of activation during training as detailed in the manuscript, not the precision of the weight.
>
> **Table A.2. Test accuracy, activation memory (ACT mem) with compression, and memory for calculating sensitivity (sens mem) in fine-tuning LLaMA2-13B and LLaMA-30B on the LIMA dataset. The micro batch size is set to 2.**
>
> | Model       | PEFT  | Scheme   | Precision | Act mem            | MMLU     | Arc-c    | PIQA     | Hellaswag | WinoGrande | BoolQ    | TruthfulQA |
> |-------------|-------|----------|-----------|--------------------|----------|----------|----------|-----------|------------|----------|------------|
> | LLaMA2-13B  | QLoRA | Baesline | 16-bit    | 8.2 GB             | **55.0** | **50.3** |   80.0   |  **79.2** |    71.5    |   80.9   |  **39.1**  |
> |             |       | ALAM     | 1-bit     | **0.8 GB (10.3x)** | 54.8     |   50.0   | **80.2** |    78.8   |  **71.6**  |  **81**  |    38.9    |
> | LLaMA-30B   | QLoRA | Baesline | 16-bit    | 15.8 GB            | 56.6     |   52.4   | **81.4** |  **82.1** |    75.0    | **82.7** |  **42.8**  |
> |             |       | ALAM     | 1-bit     | **1.6 GB (9.9x)**  | **56.7** | **52.7** |   80.8   |    81.8   |  **75.3**  |   82.5   |    42.5    |
>
> Experimental results demonstrate that our ALAM (1-bit) compresses activation by 10.3× in LLaMA2-13B and 9.9× in LLaMA-30B while closely matching the baseline in accuracy for all tasks. We will include these updated experimental results in the camera-ready version if the paper is accepted.
>
> [1] Tim Dettmers, Artidoro Pagnoni, Ari Holtzman, & Luke Zettlemoyer (2023). QLoRA: Efficient Finetuning of Quantized LLMs. CoRR, abs/2305.14314.

---

### Meta-Review · Area_Chair_gD6G · 2023-12-11

**Metareview:**

This paper proposes an activation compressed training method for training transformer based language models. The proposed method outperforms existing general ACT methods by a large margin, which could make ACT methods practically useful for training LLMs. Weaknesses include novelty concerns.

PS: though it is not considered by making the recommendation, AC personally would like to know how the wall-clock time of the proposed method compare to widely-used rematerialization methods, as ACT methods typically do not operate as fast as they seem to be due to their high memory bandwidth requirements.

**Justification For Why Not Higher Score:**

Novelty concerns.

**Justification For Why Not Lower Score:**

The proposed method outperforms existing general ACT methods by a large margin.

---

### Decision · Program_Chairs · 2024-01-16

Accept (poster)